# Systematic Review of RCTs Assessing the Effectiveness of mHealth Interventions to Improve Statin Medication Adherence: Using the Behaviour-Change Technique Taxonomy to Identify the Techniques That Improve Adherence

**DOI:** 10.3390/healthcare9101282

**Published:** 2021-09-28

**Authors:** Zoe Bond, Tanya Scanlon, Gaby Judah

**Affiliations:** Department of Surgery & Cancer, Institute of Global Health Innovation, Imperial College London, London W2 1NY, UK; zoe.bond18@imperial.ac.uk (Z.B.); Tanya.Scanlon@dhsc.gov.uk (T.S.)

**Keywords:** medicines, adherence, behaviour change, mHealth, statins, interventions, behaviour change techniques taxonomy

## Abstract

Statin non-adherence is a common problem in the management of cardiovascular disease (CVD), increasing patient morbidity and mortality. Mobile health (mHealth) interventions may be a scalable way to improve medication adherence. The objectives of this review were to assess the literature testing mHealth interventions for statin adherence and to identify the Behaviour-Change Techniques (BCTs) employed by effective and ineffective interventions. A systematic search was conducted of randomised controlled trials (RCTs) measuring the effectiveness of mHealth interventions to improve statin adherence against standard of care in those who had been prescribed statins for the primary or secondary prevention of CVD, published in English (1 January 2000–17 July 2020). For included studies, relevant data were extracted, the BCTs used in the trial arms were coded, and a quality assessment made using the Risk of Bias 2 (RoB2) questionnaire. The search identified 17 relevant studies. Twelve studies demonstrated a significant improvement in adherence in the mHealth intervention trial arm, and five reported no impact on adherence. Automated phone messages were the mHealth delivery method most frequently used in effective interventions. Studies including more BCTs were more effective. The BCTs most frequently associated with effective interventions were “Goal setting (behaviour)”, “Instruction on how to perform a behaviour”, and “Credible source”. Other effective techniques were “Information about health consequences”, “Feedback on behaviour”, and “Social support (unspecified)”. This review found moderate, positive evidence of the effect of mHealth interventions on statin adherence. There are four primary recommendations for practitioners using mHealth interventions to improve statin adherence: use multifaceted interventions using multiple BCTs, consider automated messages as a digital delivery method from a credible source, provide instructions on taking statins, and set adherence goals with patients. Further research should assess the optimal frequency of intervention delivery and investigate the generalisability of these interventions across settings and demographics.

## 1. Introduction

The global burden of cardiovascular disease (CVD) is rising, accounting for approximately 31% of global deaths [1]. Statins are the most commonly prescribed drug for those at risk of developing CVD (primary prevention) or those with CVD (secondary prevention) and are estimated to be taken by 200 million people [2,3,4,5]. Increased adherence to statins correlates with a reduced risk of CVD events, CVD-related mortality, and all-cause mortality [6,7,8]. In addition to the negative health impact of low adherence, non-adherence also results in significant healthcare costs, with one review estimating the annual per-patient cost of non-adherence to CVD medications (due to additional medical costs, unnecessary hospitalisations, and primary care visits) to range from $3347 to $19,472 [9].

A patient is typically described as adherent if they take more than 80% of their prescribed medication doses at the appropriate time over a specified period [10,11]. Statin adherence rates have been found to be 45–50% during the first six months of taking statins [12,13,14,15,16], reducing to 25% after two years [10]. This is comparable to adherence to other long-term medications [5], though there are significant negative media perceptions of statin side effects which may impact adherence to this particular medication [17].

Various factors correlate with medication non-adherence [10,18], including socioeconomic factors (female gender, younger age), disease-related factors (certain comorbidities), patient-related or psychological factors (lack of knowledge, mistrust), therapy-related factors (higher medication dose, side effects), and health system-related factors (cost of therapy, lack of appointments) [18]. As statins manage a so-called silent risk of CVD, patients may be less likely to prioritise taking them due to lack of salient symptoms [19] and therefore not perceive the benefits of taking them [18,20].

Interventions seek to alter patient-related drivers, although it is often difficult to tell which elements have successfully directed behaviour change, as many interventions are multi-faceted and use a variety of different behavioural components or techniques (e.g., using education to improve understanding [21,22] targeting patient beliefs [23], and harnessing trust in medical professionals [24,25]). The Behaviour-Change Techniques (BCT) taxonomy can systematically identify and compare the active ingredients of interventions, which may be described in different ways in different studies [26,27]. This enables effective comparisons to be made across interventions so that recommendations can be made about which specific techniques appear most effective at changing behaviour and should be used in future interventions.

There has been a rise in mobile health (mHealth) technologies, especially interventions delivered by mobile devices, such as phones and smartwatches, to alter adherence-related behaviours [28], as these technologies are scalable and relatively low cost [29]. There is a range of literature summarising the effectiveness of mHealth interventions and medication adherence for CVD [30,31,32,33,34,35,36,37,38,39]. However, only one study has focused solely on statin adherence, though this also included interventions other than mHealth interventions [39]. This review by van Driel et al. found mixed evidence supporting the link between the use of various mHealth interventions and improved adherence, though it identified a positive impact from electronic reminders [39]. As that review was conducted in 2016, and there has been an increase in the use of mHealth interventions since then, an update is necessary. Other systematic reviews, whilst demonstrating a positive or mixed impact of mHealth technologies on adherence, only focused on a single aspect of mHealth, e.g., mobile phones [33,37,38], text messaging [30], or the use of applications [32].

This review systematically compiles and assesses the available literature on the effectiveness of mHealth interventions to improve statin adherence. It builds on van Driel’s review [39] and the other reviews of mHealth interventions [30,31,32,33,34,35,36,37,38] by using the BCT taxonomy to identify the elements of interventions that effectively improve adherence and subsequently determine which BCTs are the most effective [26,27]. Two recent reviews assessed the effectiveness of mobile applications to improve medication adherence across a range of conditions using the BCT taxonomy [40,41]. Pfaeffli Dale conducted a systematic review in 2015 looking at the effectiveness of mHealth interventions in CVD medication adherence using the BCT taxonomy [36]. The current review builds on this existing research by including studies published since 2015 and focusing specifically on statin adherence, as it is the most widely prescribed medication for CVD and one that follows a typical dosing and administration form (tablets once daily).

The objectives of this review are to review the effectiveness of mHealth interventions on statin adherence, as evaluated by Randomised Controlled Trials (RCTs), and to identify BCTs employed by mHealth interventions to improve statin adherence.

## 2. Materials and Methods

The systematic review was conducted according to the Preferred Reporting Items for Systematic reviews and Meta-Analyses (PRISMA) statement [42] and its accompanying Explanation and Elaboration document [43].

The search included peer-reviewed papers, published in English, from 1 January 2000 to 17 July 2020. With the growth in the technology used within mHealth, this time period allowed the review to assess studies using the most relevant technologies. The eligibility criteria and search strategy was developed using the PICOS (population, interventions, comparator, outcomes, study design) framework [43]. Details of the search strategy are included in Appendix A (Table A1. Search strategy for Medline database in Ovid (search conducted 17 July 2020).

### 2.1. Participants

Studies including patients of any age who were prescribed statins in any setting for the primary or secondary prevention of CVD were eligible for inclusion. Studies where statins are used for patients with diabetes or stroke only were not included.

### 2.2. Intervention and Comparator

Studies with interventions employing health practices, primarily targeted at the patient, supported by any type of mobile devices were eligible for inclusion [28]. Interventions could comprise of multiple delivery methods, including both mHealth and non-digital elements. The comparator should be the usual standard of care.

### 2.3. Outcome

The primary outcome assessed was statin adherence, measured by any metric, over any follow up period and with any follow up completion rate.

### 2.4. Study Design

The review included RCTs that measured the impact of mHealth interventions on statin adherence against the standard of care. Studies were identified by electronically searching CINAHL, Cochrane Central Register of Controlled Trials (CENTRAL), EMBASE, MEDLINE, PsycINFO, and Web of Science. Additional studies were identified from a search of the grey literature and from the reference list of relevant reviews in this area and included studies.

Two reviewers (Z.B. and T.S.) independently screened the titles and abstracts of the studies for relevance and conflicts were resolved through discussion. Z.B. conducted a full-text review of all screened studies for inclusion, and T.S. screened 10% of these studies. The data extraction comprised study characteristics and results, quality assessment, and categorisation of interventions by BCT coding. Z.B. conducted all three stages and G.J. conducted BCT coding for 20% of the sample. Data was extracted into a customised template.

An assessment of the quality and internal validity of included trials was conducted using the Cochrane Risk of Bias tool Version 2 (RoB2) [44]. Blinding of participants and personnel was not included in the final score calculation, as mHealth interventions were unable to be blinded.

Reviewers identified and coded the behaviour change techniques used in the interventions and standard of care using the BCT taxonomy and the accompanying coding manual [27]. Conflicts in coding were resolved through discussion. Where necessary, study authors were contacted by email for further or missing information. If there was still insufficient detail or no response, the reviewers coded at the category level rather than specify individual techniques. As per the BCT training, the coders used a (+) to indicate “BCT present in all probability” or a (++) to indicate “BCT present beyond all reasonable doubt” [45]. Only the techniques directed at patient behaviour were coded. A BCT was recorded once even if it was mentioned multiple times in the intervention process. The extracted data on intervention delivery method, frequency, trial length, and BCT codes were then categorised and analysed based on the reported effectiveness of the trial interventions.

## 3. Results

### 3.1. Study Selection

A total of 3597 studies were identified from the systematic literature search, and 504 duplicate records (14.0%) were removed to include 3093 in the initial screening (Figure 1). Two reviewers screened the title and abstract, identified 2886 studies as irrelevant, and discussed 446 records where their initial assessments differed for inclusion in the full-text review. A total of 207 full texts were reviewed for their eligibility.

Seventeen trials described in twenty articles were screened as eligible following the full-text review [46,47,48,49,50,51,52,53,54,55,56,57,58,59,60,61,62]. The three primary reasons for exclusion were studies being trial protocols (20.6%), the medication used in the study not specified (17.4%), and ineligible interventions (14.8%). Two studies from Park (2013, 2015), Reddy (2016, 2017), and Salisbury (2016, 2017) were identified as using the same trial data. The Park 2013 [61], Reddy 2016 [48], and Salisbury 2016 [49] papers were selected, as these were the first papers published with the trial data.

### 3.2. Study Characteristics

A summary of study characteristics is shown in Table 1. The majority of studies were published after 2012. The studies were mainly conducted in high-income countries in North America and Europe. Most trials were in primary care settings, particularly in local pharmacies; however, one was conducted from hospital settings [50].

#### 3.2.1. Inclusion Criteria

Patients were primarily recruited from hospital or primary care settings. Descriptions of disease characteristics in the study inclusion criteria included diagnoses of cardiovascular disease, coronary artery or heart disease, acute coronary syndrome, and myocardial infarction. In five studies, these criteria also required participant hospitalisation [53,55,57,58,61]. Five studies required participants to be demonstrably non-adherent to statins prior to the trial [46,48,52,59,60], whilst two required participants to be newly prescribed statins in the year before the study [47,51].

#### 3.2.2. Exclusion Criteria

The most frequently used exclusion criteria was disease characteristics, including those unlikely to survive the follow up [49,57,60], severe mental illness [49,55,62], or other stated severe co-morbidities [48,49,53,55,59]. In eight studies, participants were excluded due to a physical, cognitive, or technological inability to use the intervention, including those with no telephone [48,49,50,55,57,60,61,62]. In three studies, participants were excluded if they were in receipt of care (including residence in a nursing home or not being responsible for their own medication) [56,57,60]. In two studies, participants were excluded due to lack of language proficiency [49,62].

### 3.3. Effectiveness of Interventions to Improve Statin Adherence

Information on the reported effectiveness of mHealth interventions in all studies are included in Table 1. Twelve of the seventeen included studies (71%) reported a statistically significant improvement on participant adherence to statin medication for those using mHealth interventions compared to usual care, reported in Table 2 [46,47,48,50,51,55,56,57]. Of these twelve studies, three demonstrated a statistically significant improvement in adherence in only some of the participant groups, interventions, or timepoints tested [48,52,59]. The remaining five studies found no impact of mHealth interventions on adherence [53,58,60,61,62]. The trials used a variety of measurements of statin adherence, including mean adherence measured by the average proportion of days participants opened their pill bottles or filled their prescriptions [47,48,54,56,59], Proportion of Days Covered (PDC) [46,51,52,53,57], Morisky Medication Adherence Scale (MMAS) [49,50,55,61], and statin persistence and discontinuation [58,60,62]. Five studies used self-reported adherence measures to assess the outcomes [48,49,58,61,62], whilst the rest used pharmacy dispensation data or electronic devices such as pillboxes. The effect size was calculated in seven studies where the standard deviation (SD) was provided. The effect size ranged from 0.06 to 0.75. The relative improvement in adherence ranged from 2% to 63%, with five studies (29%) demonstrating less than 10% relative improvement, three studies (18%) between 10–25%, and six studies (35%) over 25% relative improvement.

### 3.4. Intervention Characteristics

The description, duration, timing, delivery, and providers involved in the mHealth interventions employed in the included studies are described in (Table 1). The majority of studies used complex interventions that were comprised of multiple delivery components (16 studies, 11 effective studies, 69%). The mean and median number of delivery methods used was three.

The percentage of studies that demonstrated significant results varied by the delivery type used in the studies, which was a mixture of mHealth and non-digital interventions (Figure 2). The most popular mHealth delivery type was automated phone messages (eight studies) followed by electronic reminder device (six studies). Non-digital interventions included printed materials (10 studies), telephone calls (seven studies), partner support from a pre-specified individual (e.g., friend or family) (four studies), non-electronic pillboxes (four studies), and in-person consultations (two studies). The least frequently used delivery types, email, web portal, microletter, and applications, were all associated with significant improvement in adherence. Of the delivery types used in more than three studies, automated phone messages, including Interactive Voice Recognition (IVR), partner support, and non-digital pillboxes, were associated with a higher percentage of studies reporting effective interventions (75%). Whereas printed materials, which were most frequently used, were only associated with 60% studies reporting a significant improvement in adherence.

Pharmacists were the most common provider involved in the delivery of the intervention (five studies, four effective studies, 80%). Other providers included lay health workers and healthcare plan staff (six studies, four effective studies, 67%), partner support (four studies, three effective studies, 75%), and healthcare professionals (one study, no effective studies, 0%). In one study, the delivery staff was unknown [62].

The delivery period for the interventions ranged from 14 days to two years after the index date; however, the majority of studies had a delivery period of six months or less (nine studies, seven effective studies, 78%), and almost all of the trials had a delivery period of twelve months or less (15 studies, 9 effective studies, 60%). Two out of four studies (50%) where the treatment period was three months or less reported a significant improvement in adherence. This is compared to seven of nine studies (78%) when the intervention period was six months or less or five of eight studies (63%) where it was longer than six months (44 weeks to 2 years).

Whilst the delivery methods differed in their frequency of use, regular daily and monthly interventions were associated with higher rates of studies reporting significant improvement in adherence (83%, six studies and 86%, seven studies, respectively) compared to 71% in all studies. Interventions that were provided every 1–2 weeks, or on a non-recurrent basis, were associated with lower rates of studies reporting a significant improvement (50%, two studies and 0%, two studies, respectively).

### 3.5. Behaviour-Change Techniques (BCTs) Used in Included Studies

A total of 96 BCTs were coded in the intervention trial arms and 12 BCTs in usual care trial arms across the included studies. This was comprised of 30 unique BCT constructs of a possible 93 in the taxonomy (and one category of a possible 16, which was coded because there was insufficient detail to identify which individual technique was used in the intervention.) There were 64 BCTs employed in mHealth interventions, 22 BCTs employed in non-digital interventions, and 10 used by both mHealth and non-digital interventions in the same study. The number of BCTs coded in each study ranged from two BCTs (56) up to eleven BCTs (46). The mean number of BCTs coded in each study was six, and the median was five. Of the intervention BCTs coded, 33 were with a (+) to indicate “BCT present in all probability” and 63 coded with a (++) to indicate “BCT present beyond all reasonable doubt”.

The BCT most frequently used in interventions were “7.1 Prompts/cues” (16 studies, 11 effective studies, 69%), followed by “5.1 Information about health consequences” (12 studies, 9 effective studies, 75%) and “12.5 Adding objects to the environment” (ten studies, six effective studies, 60%) (Figure 3). Of the BCTs coded in more than three studies, the BCTs with the highest proportion of successful interventions was “1.1 Goal setting (behaviour)” (three studies, three effective studies, 100%) and “4.1 Instruction on how to perform a behaviour” (three studies, three effective studies, 100%). In the BCTs coded in more than three studies, the next highest proportion of BCTs coded with effective interventions was “9.1 Credible source” (nine studies, seven effective studies, 78%). The lowest proportion was for “1.2 Problem solving” (six studies, four effective studies, 67%) and “2.3 Self-monitoring of behaviour” (three studies, two effective studies, 67%).

More interventions that included a greater number of BCTs were effective compared with interventions including a smaller number of techniques. In the ten trials that coded five or fewer BCTs, six had a significant improvement (60%). There was a significant improvement in six of seven trials (86%) that used 6–11 BCTs.

### 3.6. Study Quality and Risk of Bias

A summary of the RoB2 scores across the 17 studies is presented in Figure 4. Based on the overall risk of bias judgement, four studies were determined as having “some concerns”, and 13 had a “high” overall risk of bias. No included studies were deemed to have a “low” risk. “Deviations from intended intervention” had the greatest proportion of “high” risk scores. In 13 studies, there were potential or reported failures in implementing the intervention that could have affected the outcome, such as differential delivery of the intervention. Participant engagement with the intended intervention was not sufficiently assessed and accounted for in nine studies. This is a particular issue for automated interventions where it was not possible to assess participant engagement.

## 4. Discussion

### 4.1. Summary of Evidence

There was moderate, positive evidence reported in this review of the effect of mHealth interventions to improve statin adherence. Twelve of the seventeen included studies (71%) demonstrating a significant effect, whilst five (29%) reported no impact on adherence. The studies used a variety of measures of adherence, including pharmacy refill data, electronically monitored pillbox openings, and self-reported adherence. Due to this heterogeneity, it was not possible to conduct a meta-analysis on the effect size of the mHealth interventions used. However, there was a wide range of reported relative improvement in adherence (2% to 63%) and effect size (0.06 to 0.75). There were no consistent characteristics (delivery method, length or frequency of intervention, control used, BCTs employed, or method of adherence measurement) between those with a higher relative improvement in adherence (>10%) compared to those with a lower relative improvement. The evidence on effectiveness from this review builds on the consensus from the literature that mHealth interventions improve statin adherence [30,31,34,39,63,64], though the inconsistency of results accords with the mixed evidence identified by other authors [33].

Twelve different delivery methods were used across the studies, consisting of seven mHealth and five non-digital. Most interventions were complex and multifaceted, utilising more than one delivery method, including a combination of mHealth and non-mHealth aspects. The methods used in the highest proportion of effective interventions were automated phone messages including IVR, partner support, and non-digital pillboxes. The only mHealth intervention of these three was automated phone messaging systems (IVR). There was limited evidence on the use of applications and websites, as they were only used in one study each. This was surprising given the current widespread use of apps and websites, though it may reflect the slower uptake rate of these technologies in the demographic of those taking statins. This trend may shift significantly if the study is to be repeated in a few years’ time. There is not a consensus across the literature on the most effective delivery methods to improve statin adherence. Other systematic reviews identified a range of successful delivery methods including automated phone messages and reminders [39,64], SMS [36,39,64], and pharmacist-led consultations [39]. In this review, three of six (50%) studies employing text messages were effective, while other reviews identified two of three (67%) studies [36], two of two (100%) studies [39], and nine of sixteen (56%) using IVR or SMS [64].

The majority of the included studies had relatively short trial periods (similar to trials reported in other reviews). From this review, fewer studies with treatment periods of three months or less demonstrated significant findings compared with studies with intervention periods longer than three months. However, other authors found no evidence of the impact of treatment duration on effectiveness [36,64].

The proportion of effective interventions was different with different frequencies of delivery. None of the studies that employed non-recurrent interventions demonstrated an improvement in adherence. However, there was not a clear pattern that more regular interventions were associated with higher levels of effectiveness. It may be hard to infer given the small numbers, or this may come from confounding with delivery methods (e.g., pillboxes, which are an effective intervention, are used daily). Different reviews found evidence that more frequent interventions improved effectiveness [22,64] or no evidence of a relationship [36]. Further evidence would be needed to enable conclusions to be drawn across different delivery methods.

The most frequently used Behaviour-Change Techniques (BCTs) in the identified mHealth interventions were “7.1 Prompts/cues” (16 studies, 69% effective), “5.1 Information about health consequences” (12 studies, 75% effective), and “12.5 Adding objects to the environment” (ten studies, 60% effective). In BCTs coded in more than three studies, the BCTs with the highest proportion of effective interventions were “1.1 Goal setting (behaviour)” (three studies), “4.1 Instruction on how to perform a behaviour” (three studies), and “9.1 Credible source” (nine studies), which had 100%, 100%, and 78% effective interventions, respectively. Following these, the next highest were “5.1 Information about health consequences” (12 studies), “2.2 Feedback on behaviour” (eight studies), and “3.1 Social support (unspecified)” (four studies), which all had 75% of studies with effective interventions. In this review, the BCT “7.1 Prompts/cues” was associated with 69% effective interventions, suggesting a relationship with improved effectiveness, though Kassavou found it was not associated with a larger effect size [64]. Interestingly, both the delivery method of partner support and the accompanying BCT of “3.1 Social support (unspecified)” were correlated with effective interventions (four studies, three effective studies, 75%), though there is mixed evidence for this relationship in the literature [23]. However, in one of these studies, the partner-support arm alone did not report significant results; it was only when the support was combined with an alert [59]. This review found that interventions that included more BCTs were more frequently associated with effective interventions, though other reviews that investigated CVD medication adherence and mHealth interventions found no evidence for this relationship [36,64].

The internal risk of bias for all of the included studies was rated as “some concerns” or “high”. There was a common risk of studies deviating from the protocol, as they could not measure how systematically the interventions were being delivered. In addition, automated mHealth interventions did not require a participant response, and it was difficult to gage patient engagement. Though the randomisation process appeared to be of low risk in many studies, it is important to recognise the potential for selection bias. Many trials used access to and capability of using mHealth technology as part of their inclusion and exclusion criteria, restricting certain groups and reducing their external validity.

### 4.2. Strengths and Limitations

This review followed the PRISMA guidelines and used the checklist to report the appropriate information. It used the RoB2 framework to conduct suitable assessments on the quality of evidence gathered as recommended by the Cochrane Handbook. A comprehensive search strategy was developed, and papers were identified from both journals and the grey literature to identify the widest range of material possible.

The heterogeneity of approaches and recording of outcome measures in the included studies meant a meta-analysis was not appropriate. In some studies, the interventions were developed to improve general behaviours, such as physical activity or CV risk reduction, rather than purely medication adherence (though only the BCTs addressing adherence were coded). This intervention complexity may have reduced the impact of the intervention to improve adherence. In contrast, self-reported adherence measures may have overestimated the effect. In some studies, adherence rates of over 90% were reported in the usual care group, whilst the literature evaluates the true rate to be around 50%. This review, as others have done, included adherence measures for both the process of individuals filling prescriptions and for taking medication. However, it is plausible that optimal delivery methods and BCTs may differ between these two distinct behaviours.

The BCT analysis suffered from a lack of detail in the included studies, and approximately one-third of the BCTs were coded with lower confidence level. This has implications for the reproducibility and validity of the results. This review used a stringent method of coding BCTs and only recorded them when there was sufficient information provided. Whilst this may have strengthened the assessment validity, it may have underestimated the BCTs used. As with similar reviews, there was particularly limited information on usual care and the associated BCTs may have been underreported. Finally, although a BCT may have been used in multiple delivery arms within an intervention, it was only recorded once, which may have underestimated the impact of a BCT on individual behaviour.

### 4.3. Implications for Research and Practice

Statin non-adherence is a widespread problem, and there could be vast improvements in health outcomes and cost savings using mHealth interventions, which effectively improve adherence. mHealth interventions are likely to be scalable and cost-effective delivery mechanisms; however, it is important that they use intervention content that is most likely to be effective. This research indicates the delivery methods and BCTs employed in effective interventions that can be applied by healthcare professionals and policymakers to develop interventions that are more effective at improving patient adherence to statins adherence. The four primary recommendations from this review are for practitioners to use multifaceted interventions using multiple BCTs, consider automated messages as a digital delivery method from a credible source, provide instructions on taking statins, and set adherence goals with patients. Providers might also consider providing information about health consequences, feedback on adherence behaviour, and enlisting partner support. High proportions of interventions using these techniques were effective; however, further research into the optimal delivery frequency and treatment duration for each intervention mode is required. As part of a multifaceted intervention, developers and practitioners should consider combining automated messaging with non-digital delivery methods, such as partner support and pillboxes. Where possible, this should be delivered by healthcare professionals, such as pharmacists, doctors, or nurses. However, the inconsistency of results across studies may impact the generalisability of these recommendations across settings. These recommendations were identified from studies conducted in high- and middle-income countries with well-developed health systems. Caution should be taken to recognise the potential limited reproducibility of these results in other settings, particularly where access to statins or pharmacies or familiarity with mHealth technologies may be limited. In order to be able to learn from intervention design research, the intervention components should be described in more detail within journal articles. The Behaviour-Change Techniques Taxonomy [26,27] gives a recognised and consistent way of characterising the active ingredients of an intervention.

Although a suggested advantage of mHealth interventions is their relative cost, there was no information on the cost-effectiveness of the interventions included in this review. An understanding of this is important for the adoption and diffusion of these interventions amongst patients and practitioners.

## 5. Conclusions

Whilst the prevalence of CVD across the globe continues to rise, there is a growing role for statins as the primary medication for CV prevention and treatment. There are serious detrimental health impacts of poor adherence, yet methods to improve adherence are not well established. This review found positive evidence that mHealth interventions can improve statin adherence. The methods most related to effective interventions were automated phone messages, partner support, and non-digital pillboxes. Studies that employed more Behaviour-Change Techniques (BCTs) were correlated with higher effectiveness. The BCTs most frequently associated with effective interventions were “1.1 Goal setting (behaviour)”, “4.1 Instruction on how to perform a behaviour”, and “9.1 Credible source”. Other effective techniques were “5.1 Information about health consequences”, “2.2 Feedback on behaviour”, and “3.1 Social support (unspecified)”. Based on the results of this review, the four primary recommendations from this review are for practitioners to use multifaceted interventions using multiple BCTs, consider automated messages as a digital delivery method from a credible source, provide instructions on taking statins, and set adherence goals with patients. Further research should build on this review by assessing the optimal frequency of intervention delivery and understanding the generalisability of these suggested intervention techniques across settings and demographics.

## Figures and Tables

**Figure 1 healthcare-09-01282-f001:**
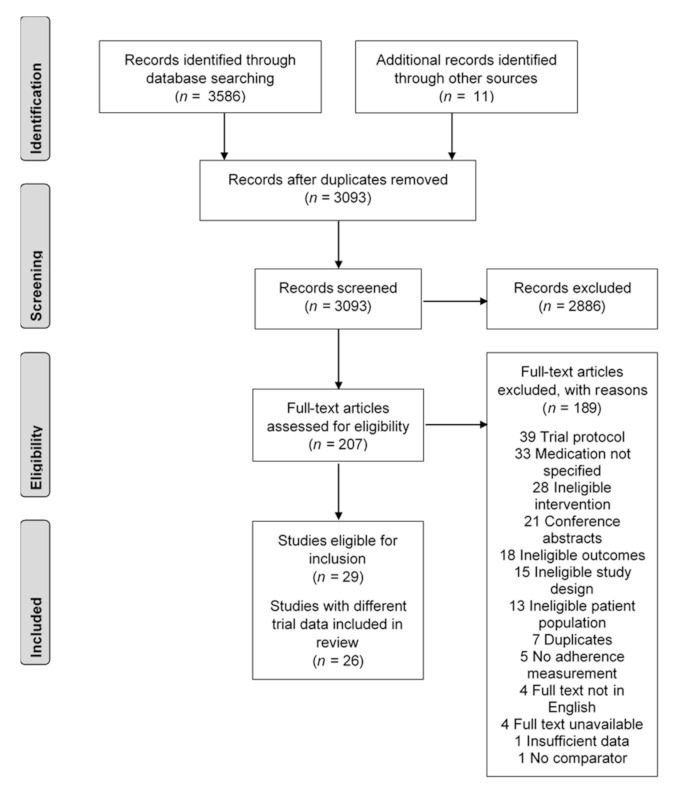
Study selection flowchart, based on Preferred Reporting Items for Systematic Reviews and Meta-Analyses (PRISMA) guidelines.

**Figure 2 healthcare-09-01282-f002:**
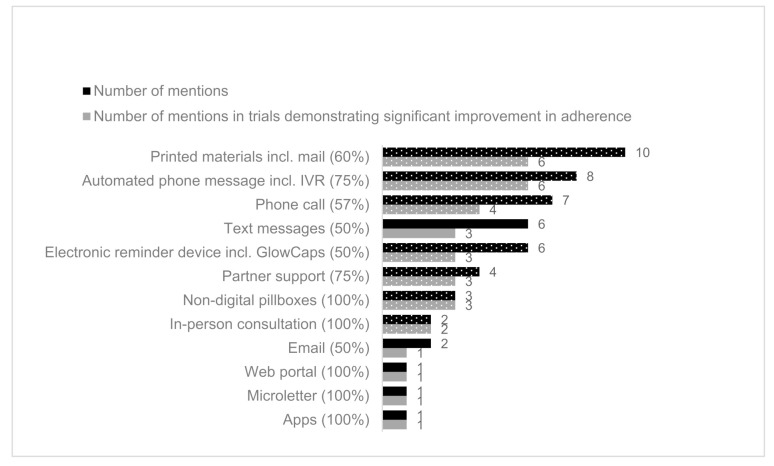
Number of mentions of delivery types in included studies (black) with effectiveness of intervention (grey). The proportion of delivery methods coded in effective interventions is provided in brackets after the delivery method name. Non mHealth delivery methods are shown in dotted. In Kessler 2018, two trial arms (alert and alert/partner) were effective, and the partner-only trial arm was not effective. For this chart, the delivery intervention “Partner” was still scored as effective for this study [59].

**Figure 3 healthcare-09-01282-f003:**
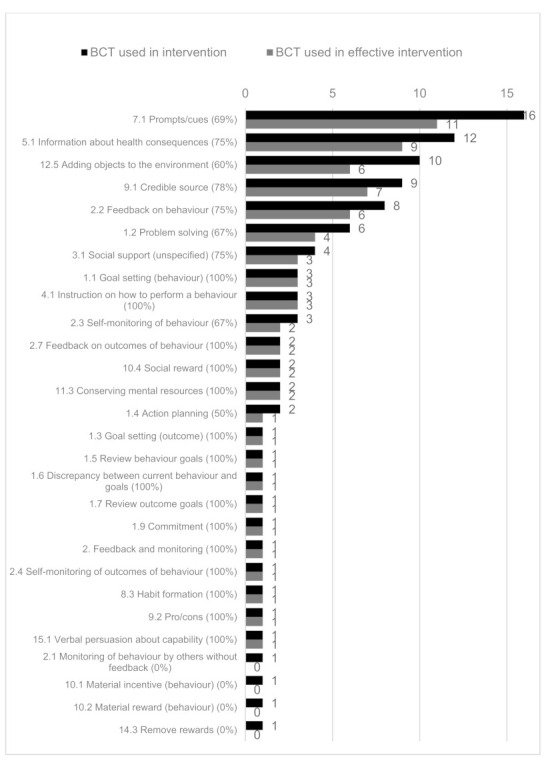
The number of Behaviour-Change Techniques (BCTs) used in all interventions (shown in black) and in effective interventions (shown in grey). The proportion of BCTs coded in effective interventions is provided in brackets after the BCT code name.

**Figure 4 healthcare-09-01282-f004:**
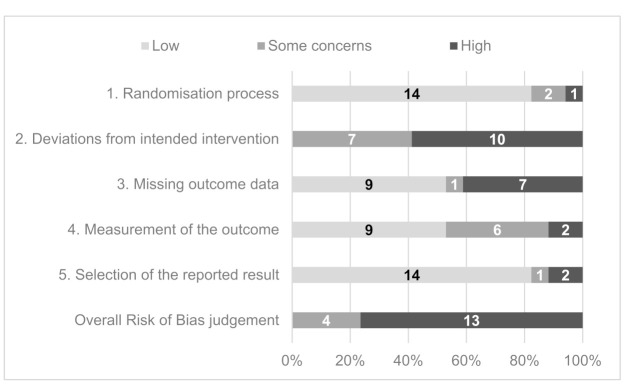
Summary Risk of Bias 2 questionnaire domain and overall scores for included studies.

**Table 1 healthcare-09-01282-t001:** Summary characteristics of included studies including author, setting, participant information, trial arms, adherence outcome data, Behaviour-Change Technique (BCT) used in the mobile health (mHealth) interventions, and overall risk of bias score from Risk of Bias 2 questionnaire.

**Author, Setting** (*shown in **bold** if sig. Improvement in Adherence*)	**Population Characteristics** (*n, Age of Eligible Participants, Average Age, % of Male Participants, Eligibility Criteria*)	**Trial Arms, Overall Risk of Bias (RoB) Score**	**Intervention Description, Delivery Method (Bold), Intervention Provider** (*Italic*)	**Adherence** (SR = Self-Reported)	**BCTs used in mHealth Intervention**
***Choudhry* et al**. ***2018*** * [46] Boston, USA	***n* = 4078, age 18–85, 60 y, 55% male** Diagnosis of hypertension, hyperlipidaemia, or diabetes and evidence of worsening disease control Non-adherent to statins	1. Pharmacist intervention 2. Usual care RoB: Some concerns	**Telephone calls** with *clinical pharmacist* based on semi structured guide tailored to patient’s activation level and identified adherence barriers Daily or weekly **text messages** **Non-digital pillboxes** **Mailed progress reports** at 6 and 9 months	Mean adherence over 12 months (SD) 42.7% (33.4) intervention 35.9% (33.0) usual care Absolute difference 8.5% (5.4–11.7% CI) unadjusted 7.6% (4.1–11.1% CI) adjusted	1.1 Goal setting (behaviour) 1.2 Problem solving 1.4 Action planning 5.1 Information about health consequences 7.1 Prompts/cues 9.1 Credible source 10.4 Social reward
***Derose**et al. 2013 **** [47] California, USA	***n* = 522, age ≥ 24, 56 y, 49% male** Evidence of poor or worsening disease control Newly described a statin and have not filled them in 1 to 2 weeks after prescription >1 yr membership in health plan	1. Automated outreach intervention 2. Usual care RoB: High	**Automated phone message** to retrieve a personalised message from the *healthcare plan staff* 1–2 weeks after prescription if missed, if unanswered messages left on machines. Two more call backs attempts if missed. If still unanswered, **letters** sent 9–11 days after initial outreach. Message states statin prescribed by clinician, importance of drug, and provides number of local health plan pharmacy.	Medication dispensed, 25 days after randomisation (%) 42.3% intervention 26.0% control Relative risk for dispensed medication 1.63 intervention vs control group (1.50–1.76 95% CI, *p* ≤ 0.001)	5.1 Information about health consequences. 7.1 Prompts/cues
***Fang and Li 2015 **** [55] Chengdu City, China	***n* = 280, adults, 54 y, 70% male** Coronary artery disease diagnosis treated in General Medicine Department	1. Text message 2. Text message + Microletter (ML) 3. Usual care (phone) RoB: High	(a) **Text message** group received medication reminders and educational material. (b) **Text message + MicroLetter** group received additional educational materials by *nurse/doctor* including disease-related information; patients can ask questions. (c) Usual care received **telephone call** to remind of medication schedule and appointments.	**SR:** Odds ratio of adherence compared to control at 6 months SMS + ML = 0.069 (95% CI, 0.032–0.151) SMS = 0.339 (95% CI, 0.183–0.629)	4.1 Instruction on how to perform a behaviour 5.1 Information about health consequences. 7.1 Prompts/cues 9.1 Credible source
***Harrison* et al. *2016 **** [56] California, USA	***n* = 41,711, age ≥ 24, 61 y, 53% male** Patients in the Kaiser Permanente Cardiovascular Disease registry (composed from diabetes, atherosclerotic CVD, heart failure, and chronic failure registries. Prescription for statin was 2–6 weeks overdue for refill	1. Automated telephone messaging system 2. Usual care RoB: High	**Automated telephone** messaging system from *healthcare plan*—delivered to live person or voicemail system. Instructed member to order a refill of their prescription by calling number or using online system.	Refilled prescription within 2 wks. after intervention 30.3% intervention 24.9% usual care *p* < 0.0001 Time from intervention to first refill (median days) 29 d intervention 36 d usual care *p* < 0.0001 Time from first refill to second refill (median days) 118 d intervention 115 d usual care *p* < 0.0001	7.1 Prompts/cues 9.1 Credible source
***Ho* et al. *2014 **** [57] Colorado, Washington, North Carolina, Arkansas, USA	***n* = 253, not provided, 64 y, 98% male** Acute Coronary Syndrome (myocardial infarction or unstable angina) as primary reason for hospital admission	1. Multifaceted intervention 2. Usual care RoB: Some concerns	(a) Medication reconciliation and tailoring. Addressed problems, adverse effects, adherence issues, synchronised prescription refill times after 1 month, answered questions, and emphasised importance of adherence. **In person** or via **telephone call**. Provided **non**-**digital pillbox** for those without one. (b) Patient education. Provided at discharge in person and in pharmacist interactions via telephone call. One week and 1 month following discharge, further info. (c) Collaborative care. Pharmacist notified primary care clinician and/or cardiologist about adherence intervention. (d) **Automated phone** messaging Reminder calls made monthly. Refill calls synchronised to 14 days before, 7 days before, on refill due date.	Composite adherence at 12 mo. (PDC > 0.8) 93.2% intervention 71.3% usual care *p* < 0.001 Average composite PDC (mean, SD) 0.95 (0.12) intervention 0.84 (0.21) usual care *p* < 0.001	5.1 Information about health consequences 7.1 Prompts/cues
*Ivers* et al. *2020* [58] Ontario, Canada	***n* = 2632, adults, 67 y, 70% male** Had coronary angiogram after myocardial infarction with evidence of obstructive artery disease and discharged from cardiac centre after procedure	1. Mail-outs 2. Mail-outs plus automated phone calls 3. Usual care RoB: High	(a) **Mailed booklets** encouraged participants in rehabilitation and long-term adherence to cardiac drugs. First two booklets enclosed a letter to take to the doctor (b) **Automated phone calls** system one to two weeks after each mail-out. If could not be contacted through this system, received **telephone call** by a trained *lay health worker*.	**SR:** Odds ratio vs usual care at 12 mo. Statin adherence in the past 7 days 1.02 (0.78–1.32) Mailouts *p* = 0.91 0.95 (0.68–1.10) Mail-outs/calls *p* = 0.73 Persistence with statins 1.00 (0.72–1.40) Mailouts *p* = 0.99 1.00 (0.75–1.32) Mail-outs/calls *p* = 0.99 Adherence to statins (PDC < 0.8) 0.89 (0.69–1.16) Mail-outs *p* = 0.39 1.04 (0.75–1.30) Mail-outs/calls *p* = 0.78	1.2 Problem solving 7.1 Prompts/cues
***Kessler* et al. *2018* †** [59] Philadelphia, USA	***n* = 179, age ≥ 18, 52 y, 65% male** CVS Health employees or their dependents with active CVS Caremark prescription coverage Excluding diabetes patients Non-adherent to statins (Medication Possession Ratio <80%)	1. Partner 2. Alert 3. Alert and Partner 4. Usual care RoB: High	(a) Friend and family acting as medication adherence ***partner*** (b) **Electronic reminder device** (wireless pill bottle) with automated message sent to individual via **email, text, or automated phone call (or multiple)** (c) Wireless pill bottle with automated message. Individual and partner both received alerts.	Overall 6 mo. average 36.0% Usual care 52.9% Alert (*p* = 0.002) 43.2% Partner (*p* = 0.25) 54.5% Alert and Partner (*p* = 0.003) Daily adherence (Odds ratio vs usual care, unadjusted) 2.75 Alert (*p* = 0.001) 1.53 Partner (*p* = 0.23) 2.92 Alert and Partner (*p* = 0.002)	2.2 Feedback on behaviour 3.1 Social support (unspecified) 7.1 Prompts/cues 12.5 Adding objects to the environment
*Kooy* et al. *2013* [60] The Netherlands	***n* = 381, not provided, 73 y, 61% male** Non-adherent patients taking statins (refill rate between 50–80%) Started statins >1 year prior to inclusion	1. Counselling with electronic reminder device (ERD) 2. ERD only 3. Usual care RoB: High	(a) Counselling session by *pharmacists* via **telephone call**: received feedback on data, asked if they were aware they were non-adherent and reasons for this, informed about benefits of statin use, received an **electronic reminder device**, informed they would be invited for follow up after one year. Ten-min counselling session made 14 days after written invitation. (b) ERD beeps until patient switches it off. It beeps every day at same time, patient can adjust the time	No. of adherent subjects at 360 days 83% usual care 90% counselling with ERD 89% ERD only *p* > 0.05	7.1 Prompts/cues 9.1 Credible source 12.5 Adding objects to the environment
*Park* et al. *2013* [61] California, USA	***n* = 90, age ≥ 21, 58 y, 77% male** History of myocardial infarction and/or Percutaneous Coronary Intervention and taking antiplatelet and statin medications	1. Text message (TM) Reminders and Education 2. TM Education Alone 3. Usual care RoB: High	(a) Received **text messages** for medication reminders and health education. Patients could select when they receive these reminders and required patients to confirm receipt. (b) Received text messages for health education on cardiovascular risk reduction. All patients received electronic pill bottles as **electronic reminder devices**	Mean doses taken at 30 days 27.7 TM Reminders/Education 27.1 TM Education Alone 25.0 usual care *p* = 0.28 Percent doses taken 92.4% TM Reminders/Education 90.1% TM Education Alone 83.3% usual care *p* = 0.28 SR: MMAS-8 at 30 days 6.43 ± 1.22 TM Rem/Ed 6.73 ± 1.49 TM Ed 6.96 ± 1.44 usual care *p* = 0.37	2.1 Monitoring of behaviour by others without feedback 5.1 Information about health consequences 7.1 Prompts/cues 12.5 Adding objects to the environment
*Párraga-Mártinez* et al. *2017* [62] Castile-La Mancha, Aragon, Galicia, Spain	***n* = 358, age ≥ 18, 59 y, 44% male** Diagnosed with hypercholesterolemia whether receiving prior therapy or not	1. Multifaceted intervention 2. Usual care RoB: High	Intervention patients received **written information** on disease and its treatment and self-completed registration cards on adherence. **Text messages** with summaries of recommendations, reminders of appointments, and **in-person consultations.**	**SR:** Adherence at 1 year 78.5% intervention 64.9% usual care *p* = 0.025 **SR:** Adherence at 2 years 77.2% intervention 64.1% usual care *p* = 0.029	7.1 Prompts/cues
***Reddy* et al. *2016* †** [48] Philadelphia, USA	***n* = 126, age 30–75, 65 y, 96% male** Veteran patients with diagnosis of coronary artery disease with documented poor adherence to statin therapy	1. Glowcap and partner feedback 2. Glowcap with individual feedback 3. Usual care RoB: Some concerns	(a) Received **electronic reminder device** GlowCap with alarm activated and weekly adherence **feedback printed report** to ***partner***. GlowCap bottle changes colour 1 h before time to take medication. If not taken, it flashes and sounds alarm. (b) Received GlowCap with alarm activated and weekly adherence feedback report to individual (c) All patients received GlowCap and educational materials on importance of adherence to statins. GlowCap features not activated in usual care group.	Adherence rate (0–3 months) 0.86 Partner feedback (*p* = 0.001) 0.89 Individual feedback (*p* < 0.001) 0.67 Usual care Adherence rate (4–6 months) 0.52 Partner feedback (*p* = 0.95) 0.60 Individual feedback (*p* = 0.75) 0.54 Usual care	1.6 Discrepancy between current behaviour and goal 2.2 Feedback on behaviour 3.1 Social support (unspecified) 7.1 Prompts/cues 10.4 Social reward 12.5 Adding objects to the environment
***Salisbury* et al. *2016 **** [49] Bristol, Sheffield, Southampton, UK	***n* = 641, age 40–74, 68 y, 48% male** Patients with >1 modifiable risk factor and QRISK2 score of a cardiovascular event in next 10 years of ≥20%	1. Cardiovascular disease risk intervention 2. Usual care RoB: High	Multifaceted intervention including regular **telephone calls** from a *lay health worker* supported by tailored algorithms and standardised scripts tailored to participants needs/goals. Linked advisors to online resources and applications to support management, which were sent to patients. Provided access to **internet portal** to monitor behaviour and outcomes. (Two-third of patients experienced some disruption over 2 months caused by provider switch)	**SR:** MMAS-4 at 12 months 3.8 intervention 3.6 usual care *p* = 0.005	1.1 Goal setting (behaviour) 1.3 Goal setting (outcome) 1.5 Review behaviour goals 1.7 Review outcome goals 2.3 Self-monitoring of behaviour 2.4 Self-monitoring of outcomes of behaviour 5.1 Information about health consequences 9.1 Credible source
***Santo* et al. *2018 **** [50] Sydney, Australia	***n* = 166, age ≥ 18, 58 y, 87% male** Patients with Coronary Heart Disease	1. Basic medication reminder app 2. Advanced medication reminder app 3. Usual care RoB: High	(a) **Basic app** provided simple daily reminders to prompt participants to take medications. (b) **Advanced app** provided interactivity including daily reminders, default settings, refill reminders, adherence stats, ability to export info, and alert ***partners***. Apps available on Australian iTunes and Google app stores	**SR:** Mean MMAS-8 at 3 months 7.11 app user (basic/advanced) 6.63 usual care *p* = 0.008 **SR:** Mean MMAS-8 at 3 months 7.19 basic app group 7.02 advanced app group 6.63 usual care *p* = 0.023	2.2 Feedback on behaviour 3.1 Social support (unspecified) 7.1 Prompts/cues
***Stacy* et al. *2009 **** [51] USA	***n* = 578, age ≥ 21, 55 y, 38% male** Patients newly prescribed statins and members of large health benefits plan	1. Experimental group 2. Enhanced care control group RoB: High	(a) Experimental group who received up to 3 separate tailored behavioural support interactions delivered **via Automated phone messages including Interactive Voice Recognition (IVR)** and **printed material** (b) Enhanced care control groups received non-tailored behavioural advice from a single IVR call with a generic guide in the mail. Calls referred to respondents to the health place **internet** site for additional information.	6-mo. point prevalence persistency 70.4% intervention 60.7% enhanced care *p* < 0.05 Continuous persistence 52.5% experimental 44.3% enhanced care *p* < 0.10 MPR ≥ 80% 47.0% experimental 38.9% enhanced care *p* < 0.10	1.1 Goal setting (behaviour)1.2 Problem solving 1.9 Commitment 2. Feedback and monitoring 5.1 Information about health consequences 7.1 Prompts/cues 8.3 Habit formation 9.2 Pros/cons 15.1 Verbal persuasion about capability
***Vollmer* et al. *2014*****†** [52] Northwest, Hawaii and Georgia, USA	***n* = 16,380, age ≥ 40, 64 y, 54% male** Statin users from Kaiser Permanente regions who were nonadherent <90% to treatment	1. Interactive Voice Recognition (IVR) 2. Enhanced Interactive Voice Recognition (IVR+) 3. Usual care RoB: Some concerns	(a) IVR participants received **automated phone calls** when refill due/overdue and to educate patients and help them refill prescriptions (separate calls). Both call types offered a transfer to automated pharmacy refill line. Accompanied with mailed **printed materials**. (b) IVR+ participants also received personalised reminder letter if 60–89 d overdue and a live **telephone call** from local *pharmacy staff* if they were ≥90 d overdue as well as EMR-based feedback to their primary care provider. Received personalised health report, **non-digital pillbox**, and bimonthly mailings.	Statin adherence at 12 months 0.57 IVR 0.58 IVR+ 0.55 Usual care p-value for IVR/IVR+ vs UC < 0.000 Statin users with ≥80% adherence 35.9% IVR 35.8% IVR+ 32.9% Usual care p-value for IVR/IVR+ vs UC < 0.002 This relationship is not stat significant in those with adherence <0.4 at start of trial	5.1 Information about health consequences 7.1 Prompts/cues 9.1 Credible source
*Volpp* et al. *2017* [53] Pennsylvania, USA	***n* = 1509, age 18–80, 61 y, 66% male** Patients with Acute Myocardial Infarction immediately post-hospitalisation and currently prescribed >2 secondary prevention medications	1. Multifaceted intervention 2. Usual care RoB: High	Intervention included provision of up to 4 **electronic reminder devices** (Vitality GlowCaps or MedSignal device). Assignment of an engagement advisor who would attempt to contact patients via **telephone call** or **mail letter** they had not opened device in 6 days. Enlisting a family member or friend as a support partner. Engagement incentives that will use lotteries dependent on adherence. Self-service/customisation of Way to Health platform communication methods including **text message, automated phone message, email**.	PDC (strict definition) at 12 months 0.72 intervention 0.69 usual care *p* = 0.23 PDC (intermediate def.) at 12 months 0.80 intervention 0.78 usual care *p* = 0.26 PDC (relaxed def.) at 12 months 0.83 intervention 0.81 usual care *p* = 0.27	1.2 Problem solving 2.2 Feedback on behaviour 3.1 Social support (unspecified) 7.1 Prompts/cues 10.1 Material incentive (behaviour) 10.2 Material reward (behaviour) 12.5 Adding objects to the environment14.3 Remove rewards
***Vrijens* et al. *2006 **** [54] Flanders, Wallonia, Belgium	***n* = 429, age ≥ 18, 62 y, 52% male** Patients who have been taking atorvastatin for at least three months	1. Pharmaceutical care program 2. Usual care RoB: High	**In-person consultation** with the patient’s *pharmacist* reviewing the electronically compiled dosing history and discussing educational message and provided **printed materials**. An **electronic reminder device** beep-card that reminds patient of dosing time.	Adherence after 90 days 96.43% intervention 94.33% usual care *p* = 0.003 Adherence after 300 days 95.89% intervention 89.37% usual care *p* < 0.001	2.2 Feedback on behaviour 2.3 Self-monitoring of behaviour 7.1 Prompts/cues 9.1 Credible source 12.5 Adding objects to the environment

Abbreviations: Electronic Reminder Device (ERD), Interactive Voice Recognition (IVR), Morisky Medication Adherence Scale (MMAS), Medication Possession Ratio (MPR), Proportion of Days Covered (PDC), Self-reported (SR), Standard Deviation (SD), Text Message (TM). Author names in **bold *** represent those which demonstrated a significant improvement in adherence, names in **Bold †** represent those with a significant improvement but not in all subgroups.

**Table 2 healthcare-09-01282-t002:** Adherence rates reported in trials reporting statistically significant improvement in adherence, including adherence rate in usual care and intervention trial arms, the standard deviation (SD) of the usual care arm adherence if provided, effect size, and relative improvement in adherence.

	Adherence Measurement		Usual Care	Intervention	SD	Effect Size	Relative Improvement
**Choudhry ***	Mean PDC over 12 months		36%	46%	36%	0.28	28%
**Derose ***	% participants who had medication dispensed		26%	42%			63%
**Fang and Li ***	MMAS-4	SMS/ML		−2.674	−6.71	0.40	
		SMS		−1.082	−3.43	0.32	
**Harrison ***	% participants who had filled prescription		25%	30%			22%
**Ho ***	PDC > 80%		71%	93%			31%
	Mean PDC		84%	95%	21%	0.52	13%
**Kessler †**	Mean % pill bottle openings in 6 months	Partner + alert	36%	55%	25%	0.75	51%
		Alert	36%	53%	25%	0.69	47%
**Reddy †**	% pill bottle openings	Partner feedback	67%	89%			33%
		Individual feedback	67%	86%			28%
**Salisbury ***	MMAS-4		3.6	3.8	0.8	0.25	6%
**Santo ***	MMAS-8		6.63	7.11			7%
**Stacy ***	PDC > 80%		61%	70%			16%
**Vollmer †**	Modified PDC	IVR+	55%	58%	35%	0.09	5%
		IVR	55%	57%	35%	0.06	4%
**Vrijens ***	% pill bottle openings		94%	96%			2%

Abbreviations: Interactive Voice Recognition (IVR), Microletter (ML), Morisky Medication Adherence Scale (MMAS), Proportion of Days Covered (PDC), Short Message Service (SMS). Author names in **Bold *** represent those which demonstrated a significant improvement in adherence, names in **Bold †** represent those with a significant improvement but not in all subgroups.

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
