# Peer review of "Systematic Review of RCTs Assessing the Effectiveness of mHealth Interventions to Improve Statin Medication Adherence: Using the Behaviour-Change Technique Taxonomy to Identify the Techniques That Improve Adherence"

_healthcare, 2021, doi:10.3390/healthcare9101282_

Round 1
Reviewer 1 Report
Please find my comments in the annexed file.

Reviewer 2 Report
Comments:
- In abstract section remove the numbers 1.1, 4.1(page 1, line 23); 9.1, 5.1 (page 1, line 24); and 2.2, 3.1 (page 1, line 25).
- On page 2, reference 39 (lines 73 and 86) should be written in square brackets.
- The phone devices are improved amazingly in the last decade, the development of apps is growing continuously, and almost all people are using them. The fact that the phone messages be the most important pharmacist intervention could be due to the analysis of studies previous to 2010, or that these apps are not freely available? Please discuss.
- In some studies, adherence improves slightly (2%), in most cases at 17-20%. Seems like that the interventions do not reach the same level of improvement, please enrich the discussion with these discrepancies.
